# Distinct Adipogenic and Fibrogenic Differentiation Capacities of Mesenchymal Stromal Cells from Pancreas and White Adipose Tissue

**DOI:** 10.3390/ijms23042108

**Published:** 2022-02-14

**Authors:** Heja Aga, George Soultoukis, Mandy Stadion, Francisco Garcia-Carrizo, Markus Jähnert, Pascal Gottmann, Heike Vogel, Tim J. Schulz, Annette Schürmann

**Affiliations:** 1Department of Experimental Diabetology, German Institute of Human Nutrition Potsdam-Rehbruecke (DIfE), 14558 Nuthetal, Germany; heja.aga@dife.de (H.A.); mandy.stadion@gmx.de (M.S.); markus.jaehnert@dife.de (M.J.); pascal.gottmann@dife.de (P.G.); heikevogel@dife.de (H.V.); 2German Center for Diabetes Research (DZD), München-Neuherberg, 85764 München, Germany; george.soultoukis@dife.de (G.S.); tim.schulz@dife.de (T.J.S.); 3Department of Adipocyte Development and Nutrition, German Institute of Human Nutrition Potsdam-Rehbruecke (DIfE), 14558 Nuthetal, Germany; francisco.garcia@dife.de; 4Research Group Genetics of Obesity, German Institute of Human Nutrition Potsdam-Rehbruecke (DIfE), 14558 Nuthetal, Germany; 5Research Group Molecular and Clinical Life Science of Metabolic Diseases, Faculty of Health Sciences Brandenburg, University of Potsdam, 14469 Potsdam, Germany; 6Institute of Nutritional Sciences, University of Potsdam, 14558 Nuthetal, Germany

**Keywords:** MSCs, fatty pancreas, WAT, lineage commitment, transcriptomics, miRNAs

## Abstract

Pancreatic steatosis associates with β-cell failure and may participate in the development of type-2-diabetes. Our previous studies have shown that diabetes-susceptible mice accumulate more adipocytes in the pancreas than diabetes-resistant mice. In addition, we have demonstrated that the co-culture of pancreatic islets and adipocytes affect insulin secretion. The aim of this current study was to elucidate if and to what extent pancreas-resident mesenchymal stromal cells (MSCs) with adipogenic progenitor potential differ from the corresponding stromal-type cells of the inguinal white adipose tissue (iWAT). miRNA (miRNome) and mRNA expression (transcriptome) analyses of MSCs isolated by flow cytometry of both tissues revealed 121 differentially expressed miRNAs and 1227 differentially expressed genes (DEGs). Target prediction analysis estimated 510 DEGs to be regulated by 58 differentially expressed miRNAs. Pathway analyses of DEGs and miRNA target genes showed unique transcriptional and miRNA signatures in pancreas (pMSCs) and iWAT MSCs (iwatMSCs), for instance fibrogenic and adipogenic differentiation, respectively. Accordingly, iwatMSCs revealed a higher adipogenic lineage commitment, whereas pMSCs showed an elevated fibrogenesis. As a low degree of adipogenesis was also observed in pMSCs of diabetes-susceptible mice, we conclude that the development of pancreatic steatosis has to be induced by other factors not related to cell-autonomous transcriptomic changes and miRNA-based signals.

## 1. Introduction

Obesity, defined as an excess accumulation of white adipose tissue (WAT) mass, is a global health concern that is linked to an increased risk for metabolic perturbations such as type-2-diabetes (T2D), cardiovascular diseases, and non-alcoholic fatty liver disease (NAFLD) [1]. In obesity, WAT expansion occurs through (i) lipid filling within existing fat cells (adipocyte hypertrophy) and (ii) formation of new adipocytes from mesenchymal stromal cells (MSCs) with adipogenic differentiation potential (adipocyte hyperplasia), also well-known as adipogenic precursor cells (APC) [2]. When MSCs exhibit a low ability to differentiate into mature fat-storing adipocytes, the rate of adipocyte hypertrophy is increased [2,3]. However, excessive levels of circulating lipids combined with a limited adipocyte storage capacity result in ectopic fat accumulation in non-adipose tissue organs such as liver and muscle, taking the shape of intra-hepatocyte and intra-myocyte lipid droplet stores or as bona fide mature adipocytes interspersed within bone tissue, and the pancreas [4,5,6,7,8]. These ectopic lipid depositions are linked to organ dysfunction, insulin resistance, and the predisposition to develop T2D [4,8]. Even though, the impact of NAFLD on glucose metabolism has been widely studied [9,10], the clinical significance of equivalent fat cells in the pancreas (non-alcoholic fatty pancreas disease, NAFPD) has long been underestimated. Today, there is growing evidence that NAFPD plays a substantial role in human pancreatic fibrosis, pancreatic cancer, and β-cell dysfunction [11]. 

Recently, we have reported that genetically diabetes-prone New Zealand Obese (NZO) mice, an obese mouse-strain used as model for polygenic diabetes, exhibit more adipocytes in the pancreas than diabetes-resistant, leptin-deficient B6.V-Lep ob/ob (B6-ob/ob) mice [12]. Moreover, co-culture of pancreatic islets and adipocytes derived from MSCs of the pancreas or inguinal white adipose tissue (iWAT) resulted in hypersecretion of insulin [12]. However, the role of pancreas-resident MSCs in modulating organ health is still unclear. 

To investigate the function of pancreas MSCs (pMSC) and to compare it with iWAT MSCs (iwatMSCs), we isolated a defined subpopulation of both MSCs from diabetes-resistant C57BL/6J (B6) mice using fluorescence-activated cell sorting (FACS) and analyzed their transcriptional and miRNA profiles. MSCs of both sources exhibited marked differences in mRNA and miRNA expression profiles as well as in adipogenic and fibrogenic differentiation capacity.

## 2. Results

### 2.1. Distinct Gene Expression Profiles of pMSCs and iwatMSCs

In order to compare pancreas (pMSCs) and iWAT MSCs (iwatMSCs), we utilized processed pancreases and iWAT of the same male, 8-week-old standard diet fed B6 mice and sorted for a non-endothelial (CD31-), non-hematopoietic (CD45-), adipogenic progenitor-type (SCA1+) MSC subpopulation via FACS (Figure 1A,B). Comparative transcriptome analysis of 26,735 expressed transcripts revealed 1,456 differentially expressed transcripts referring to 1227 genes (DEGs) with 757 significantly up- and 699 significantly down-regulated genes in pMSCs compared to iwatMSCs (Figure 2A). Gene ontology (GO) enrichment analysis allocated DEGs to 15 significant biological processes (gene count ≥ 10, fold enrichment > 2, unadjusted *p* < 0.05) (Figure 2B), of which most were assigned to positive regulation of developmental processes and fat cell differentiation, cell adhesion, cell migration, and others involving re-arrangement of the cytoskeleton. An enrichment of GO terms including collagen fibril organization, negative regulation of BMP signaling pathway, and negative regulation of fibroblast proliferation pointed toward a potentially alternative fibrogenic cell fate of both types of MSCs. Ingenuity pathway analysis (IPA) identified seven networks, of which one significant pathway linked 11 DEGs directly or indirectly to genes well-known for their role in cell growth and adipogenesis, e.g., *Irs-1* (insulin receptor substrates 1) and *Irs-2*, *Pparg* (peroxisome proliferator activated receptor gamma), *Lepr* (leptin receptor) or *Srebf* (sterol regulatory element-binding transcription factor). This does not only emphasize the strong association of MSCs with the adipogenic lineage but the versatile developmental nature of mesenchymal-type precursor cells (Figure 2C).

Proteins that are actively transported within the secretory pathway play a pivotal role in the regulation of critical cellular functions and are defined as “secreted proteins”. A majority of these proteins, including cytokines, growth factors, and hormones, participate in local and systemic signaling functions [13]. Therefore, an in-silico secretome analysis was performed using the Human Protein Atlas. Interestingly, 154 of the 1227 DEGs were estimated to encode for secreted proteins, of which 112 genes were higher expressed in pMSCs and 42 in iwatMSCs (Figure 3, Appendix A). To gain insight into functional processes of potentially secreted proteins, we looked at the top 10 predicted secreted proteins either abundant in pMSCs or iwatMSCs, respectively. Genes of potentially secreted proteins in iwatMSCs were mainly allocated to cell proliferation (*Prl*, *Grem2*, *Sfrp2*) and angiogenic processes (*Mmp2*, *Mmp19*, *Thbs2*). In contrast, transcripts of secretory proteins in pMSCs were assigned to proliferative events (*Igfbp4*, *Igfbp7*, *Smoc2*, *Slit3*) and immunomodulation (*Cxcl2*, *Angptl2*, *Gpx3*) (Figure 3).

### 2.2. miRNA Expression Patterns of pMSCs and iwatMSCs

miRNAs are short, 19–23 nucleotides long single-stranded RNA sequences playing a critical role in the post-transcriptional regulation of gene expression. They act in functional networks as each miRNA targets a few to several mRNAs [14]. Expression analysis of 1240 miRNAs in pMSCs and iwatMSCs revealed that 121 were differentially expressed, of which 66 were higher and 55 lower expressed in pMSCs compared to iwatMSCs (Figure 4A). To link differences in miRNA expression to altered mRNA levels, we performed a target gene prediction analysis and identified 58 differentially expressed miRNAs putatively regulating 510 genes (Appendix A). As miRNAs usually negatively regulate gene expression, only target genes were considered that fit to the direction of miRNA expression (e.g., miRNA expression up, target gene expression down). Next, GO enrichment analyses were performed with predicted target genes for pMSCs and iwatMSCs, separately (gene count ≥ 3, fold enrichment > 2, unadjusted *p* < 0.05) (Figure 4B,C). In fact, comparison of both GO-circles showed no overlaps between the depicted biological processes, indicating key differences in miRNA-regulated gene expression in both MSC types. While miRNA targets of pMSCs are predominantly involved in transcription and gene expression processes, iwatMSCs are mainly linked to cytoskeleton-related pathways and proliferation. 

Furthermore, both MSC populations differ in their miRNA expression signatures: While miR-204-5p, miR-211-5p, miR-196a-5p, and miR-196b-5p were almost exclusively expressed in iwatMSCs (Figure 4D), miR-335-5p, miR-375-3p, miR-429-3p, and miR-200a-3p were mainly detected in pMSCs (Figure 4E). For miR-204-5p we determined 41 regulated genes (Appendix A) with a higher expression in pMSCs than in iwatMSCs. The strongest repression with log2-fold changes of >1.5 were observed for those candidates shown in the box of Figure 4D. Among these suppressed genes is the ephrin receptor gene, *Ephb2*, which can also act as a ligand and regulates diverse cellular processes including motility, division, and differentiation [15]. Other targets are *Dnajc14*, a heat shock protein; *Slc7a2*, a cationic amino acid transporter; *Arhgef5*, a Rho guanine nucleotide exchange factor, pointing toward a function in controlling cytoskeletal organization. Theoretically, miR-211-5p targets three genes (*Tcf4*, *Cux1*, *Trps1*) which exhibit a lower expression in iwatMSCs than in pMSCs. However, only *Trps1*, a transcriptional repressor, reaches the threshold of >1.5-fold differential expression (see box in Figure 4D). miR-196a-5p and miR-196b-5p both target the phospholipase A2 receptor 1 (*Pla2r1*), even though the log2-fold changes here were <1.5 (Appendix A). This gene is described to promote cellular senescence by inducing DNA damage [16]. Thus, via an elevated miR-196a/b expression iwatMSCs might lower *Pla2r1* levels and protect against senescence.

The pMSC-abundant miRNA miR-335-5p, theoretically targets 32 genes which show a lower expression in comparison to iwatMSCs. Among these, 13 genes exhibit a repression with log2-fold changes of >1.5 (see box in Figure 4E). *Met* (mesenchymal endosomal transition) encodes a receptor tyrosine kinase, which is activated by hepatocyte growth factor (HGF) and thereby involved in the regulation of a variety of biological processes including cell motility, cell proliferation, the epithelial-to-mesenchymal transition, and the development and progression of cancer cells [17]. Another interesting target is *Hmgcr* (3-hydroxy-3-methylglutaryl-coenzyme A reductase), the rate-limiting enzyme for cholesterol synthesis. miR-375-3p targets *Yap1* (yes-associated protein 1) with a threshold of >1.5. This gene encodes an oncogene, a downstream nuclear effector of the Hippo signaling pathway. In its unphosphorylated state, *Yap1* translocates to the nucleus where it binds to TEAD family transcriptions factors and regulates genes promoting proliferation [18]. miR-429-3p is thought to target the glycosyltransferase *St3gal2*. Deletion of this gene induces late onset of obesity and insulin resistance, particularly in adipose tissue, which was associated with altered ganglioside profiles [19]. miR-200a-3p targets *Pafah1b1* (platelet activating factor acetylhydrolase 1b regulatory subunit 1), which causes type I lissencephaly, a neuronal migration disorder, upon haploinsufficiency [20]. In mice, genetic deletion of *Pafah1b1* in the liver attenuated lipid release and thereby increased fat accumulation in the hepatocytes [14].

### 2.3. Decreased Adipogenic Potential of pMSCs

As a large number of mRNAs and miRNAs differed in their abundance in pMSCs and iwatMSCs and depot-specific differences with regard to the adipogenic capacity are well-known [7,21], we tested if they varied in their lineage commitments by performing differentiation assays. To this end, pMSCs and iwatMSCs from B6 mice were subjected to a specific adipocyte differentiation protocol or incubated without the differentiation cocktail to evaluate the levels of spontaneous adipogenesis. The iwatMSCs showed some degree of spontaneous differentiation to adipocytes and adipogenesis was observed in almost all cells following the adipogenic induction treatment. In contrast, pMSCs exhibited a markedly decreased spontaneous as well as induced adipogenesis compared to iwatMSCs as reflected by a lower number of Oil Red O-positive cells (Figure 5A). Quantification of the lipid droplets showed an almost 70% lower adipogenic potential after adipogenesis induction in pMSCs (Figure 5B). In line with this, expression levels of adipocyte marker genes like the transcription factors *Cebpa* (CCAAT/enhancer-binding protein alpha) and *Pparg* (peroxisome proliferator-activated receptor gamma) as well as *Adipoq* (the adipokine adiponectin) and *Plin1* (lipid-droplet associated protein perilipin 1) were 4-to 9-fold lower expressed in pMSCs in comparison to iwatMSCs (Figure 5C). Remarkably, mRNA levels of *Slc2a4* (glucose transporter 4) were close to the detection limit in pMSCs.

### 2.4. Increased Fibrogenic Potential in pMSCs

As the adipogenic potential of pMSCs was significantly lower than that of iwatMSCs, we hypothesized an alternative cell fate programming for pMSCs. Our expression analysis of transcripts of putatively secreted proteins (Figure 2B) pointed toward a potential fibrogenic lineage commitment. Therefore, we tested the fibrogenic capacity by treating pMSCs and iwatMSCs with TGF-β1 followed by staining of fibrogenesis marker α-SMA. The representative pictures (Figure 6A) and the morphometric analysis displayed significantly higher α-SMA staining intensity in pMSCs (Figure 6B) supporting a higher capacity to undergo fibrogenesis compared to iwatMSCs. In agreement with this, expression levels of representative fibrogenesis marker genes like *Acta2* (actin alpha 2), *Col1a1* (collagen type1 alpha 1) and *Pdgfra* (platelet-derived growth factor receptor A) were higher expressed in pMSCs than in iwatMSCs (Figure 6C). 

To test whether the implied fibrogenic development of pMSCs affected insulin secretion, glucose-stimulated insulin secretion experiments were performed with isolated pancreatic islets of B6 mice alone or with islets that were co-cultured with fibrogenically differentiated, i.e., TGF-β1-treated, pMSCs. In fact, insulin secretion did not differ between both conditions. It was increased in response to 20 mM glucose by 6–7-fold in the presence or absence of fibrotic cells. Thus, acute fibrogenesis of MSCs in the pancreas does not impair insulin secretion (Figure 6D).

### 2.5. Similar Cell-Fate Decisions in pMSCs Derived from Diabetes-Susceptible and Diabetes-Resistant Mice

As we have previously shown that diabetes-prone NZO mice exhibit increased amounts of fat cells in their pancreas, while diabetes-resistant B6-ob/ob have nearly no fat cells in this organ, we reasoned that pMSCs derived from NZO mice might show a more pronounced adipogenic cell fate commitment. Therefore, pMSCs and iwatMSCs were isolated from 8-week-old B6 and NZO mice (as described in Figure 1). When comparing the sorted cell counts normalized to tissue mass, we did not identify any significant differences between the two mouse strains (Appendix A). Next, an expansion period of 4 days was followed by the induction of adipogenesis or fibrogenesis. As before, adipogenic potential of pMSCs was limited in comparison to that of iwatMSCs derived from both B6 and NZO mice (Figure 7A,B). In line with this, pMSCs of B6 and NZO displayed an increased α-SMA signal intensity compared to iwatMSCs. Again, no strain-specific differences were found (Figure 7C,D). These findings clearly show that factors other than cell-autonomous elements of pMSCs, e.g., endocrine signals from liver or other tissues, may induce adipocyte accumulation in pancreas of diabetes-susceptible NZO mice.

## 3. Discussion

Mesenchymal stromal cells (MSCs) are multipotent stem cells present in the stroma of many tissues and can give rise to a number of cells including adipocytes [22]. Our study identified marked variations in the expression profile and differentiation characteristics between MSCs isolated from pancreas (pMSCs) and iWAT (iwatMSCs). In general, MSCs are able to differentiate into several lineages, e.g., adipocytes and fibroblasts [3,23]. As expected, iwatMSCs exhibit a major potential to differentiate into adipocytes. In contrast, despite sorting cells with the same set of surface marker proteins, pMSCs preferentially differentiated into fibrogenic rather than adipogenic cells. Considering that spontaneous adipogenic differentiation can impede the differentiation potential of preadipocytes into other lineages [24], the lower spontaneous adipogenic capacity of pMSCs compared to iwatMSCs which we observed in our functional analyses agrees with the increased fibrogenic cell-fate of pMSCs.

Interestingly, also pMSCs of diabetes-susceptible NZO mice, in which we previously detected a high number of adipocytes in pancreas, exhibit a low degree to differentiate into adipocytes [12]. However, beside NZO mice, another T2D mouse model has been described to develop fat infiltrations in the pancreas [25]. Thus, we hypothesize that other species-specific factors are responsible for the adipocyte accumulation in pancreas.

The distinctive cell lineage fates of both MSC populations are clearly reflected by the transcriptome. In pMSCs, enriched genes are linked to a function in cytoskeleton, cell growth, and particularly in negative regulation of BMP signaling and fibroblast proliferation, indicating their elevated fibrogenic cell fate. In fact, our in vitro studies showed the differentiation of pMSCs to fibrogenic cells which might indicate that pMSCs play a role in fibrosis of the pancreas, e.g., in the development of pancreatitis. Due to the expression of genes related to a positive regulation of fat cell differentiation, pMSCs are principally able to become adipocytes. The IPA (Ingenuity Pathway Analysis) detected 8 genes which are more abundant in pMSCs than in iwatMSCs, and which are organized in an adipocyte-specific network. These genes include the transcription factor *Isl1* (insulin gene enhancer binding protein 1) which was shown to be markedly but transiently upregulated in 3T3-L1 cells shortly after the initiation of differentiation. Its overexpression during early differentiation suppressed *Pparg* expression and inhibited adipogenesis [26]. *Tcf4* (transcription factor 4) is involved in Wnt signaling and forms a complex with β-catenin [27]. Interestingly, treatment of cardiac stem cells with high glucose concentrations inhibited the β-catenin/TCF4 pathway and promoted adipogenesis [28]. Therefore, it is conceivable that hyperglycaemia of NZO mice inhibits the expression of *Tcf4* in pMSCs, which could ultimately induce their differentiation to adipocytes. Transcripts of two fibroblast growth factors, *Fgf1* and *Fgf2*, are higher expressed in pMSCs than in iwatMSCs and both are involved in the regulation of fat cell differentiation. FGF1 promotes adipogenesis in preadipocytes by increasing the expression of *Pparg* [29]. In contrast, FGF2 rather inhibits adipogenesis because treatment of proliferating preadipocytes with FGF1 reduced the *Fgf2* expression markedly, leading to decreased proliferation and increased expression of adipogenic marker genes [30]. Thus, the fact that both *Fgf1* and *Fgf2* are expressed at higher levels in pMSCs might explain why their adipogenic cell fate is lower than that of iwatMSCs.

miRNAs have emerged as key epigenetic players during MSC differentiation [22]. They are potent regulators because each miRNA controls and finetunes the expression of several mRNA species. According to the expression profile of miRNAs and the respective target prediction, our data indicate that approximately 40% of differences in mRNA expression of pMSCs and iwatMSCs are mediated by miRNAs. The fact that two genes, *Jade1* and *Ubr4*, were predicted as targets for the iwatMSC-specific miR-204-5p and the pMSC-specific miR-335-5p, caught our attention. In both cases, two different transcripts were supposed to be affected by the miRNAs. As the resulting protein sequences did not (JADE1) or only moderately (UBR4) differ, their levels and function should be similar in both MSC types.

Interestingly, among the differentially regulated miRNAs, several candidates are known to be involved in adipogenesis, like miR-27-3p, for which we identified 74 targets. These include transcription factors like *Arnt2* (aryl hydrocarbon receptor nuclear translocator 2) and *Nr4a2* (nuclear receptor subfamily 4 group A member 2) and the fatty acid transporter *Slc27a1* (also designated FATP1). Interestingly, miR-143-3p, which was the first reported miRNA to be involved in adipogenesis [31], exhibits higher expression in pMSCs and targets *Igfbp5* (insulin-like growth factor-binding protein 5) and *Hmgcr* (HMG-CoA reductase involved in cholesterol synthesis). 

Among the differentially expressed miRNAs is also miR-31-5p, which we identified in an obesity locus on chromosome 4and to affect expression of genes relevant for glucose transport and insulin signaling [32]. miR-31-5p was shown to be upregulated by BMP2 in MSCs and to inhibit adipogenesis [33]. Interestingly, the miRNA-335-5p, highly abundant in pMSCs, regulates, among other genes, *Cav1* (caveolin 1). As inhibition of *Cav1* impairs adipogenesis of bone marrow-derived MSCs [34], it can be hypothesized that adipocyte differentiation of pMSCs, beside other mechanisms, is inhibited via miR-335-*Cav1* interaction.

Furthermore, also several of the 58 regulated miRNAs were previously associated with pro-fibrotic events. For instance, anti-fibrotic miRNAs miR-221-3p [35], miR-196a-5p [36], and miR-20a-3p [37] were lower expressed in pMSCs and higher in iwatMSCs, while pro-fibrotic miRNAs miR-10a-5p [38] and miR-23b-3p [39] were enriched in pMSCs. More specifically, miR-10a-5p overexpression was shown to promote COL1A1, COL1A3, α-SMA, and TGF-β1 protein expression and thus increased the levels of atrial fibrillation and cardiac fibroblasts in rat models [38]. It is known that miR-23b-3p together with miR-27b-3p promotes atrial fibrosis by targeting TGFBR3 [39] and that sponging of miR-196a-5p increases the expression of the lncRNA H19 which leads to an increased fibroblast activation through COL1A1 in the human fibroblast cell line MRC-5 [36]. Therefore, the cell-specific miRNA signature might, in part, explain a fibrogenesis-favoring cell fate of pMSCs.

Several internal and external factors influence proliferation and differentiation of MSCs and these factors include exposure to a range of secreted proteins. Insights into the secretome of different MSCs species could contribute to a better understanding of respective signaling functions. We did not directly analyze the secretome of pMSCs and iwatMSCs but performed an in-silico analysis for transcripts encoding for proteins that are released. Generally, our analysis showed a higher number of genes encoding secreted protein in pMSCs (110 mRNAs of secreted proteins) compared to iwatMSCs (44 mRNAs of secreted proteins). Among the 10 secreted proteins that were predicted to be encoded by genes highly expressed in iwatMSCs, *Sfrp2* (secreted frizzled related protein 2) is a known modulator of the Wnt signaling pathway and is described to be a mediator of adipogenic and neuronal differentiation in dental tissue-derived MSCs [40]. *Mmp2* and *Mmp19* (matrix metallopeptidases 2 and 19), abundant in iwatMSCs, have a pivotal role in adipocyte differentiation, which is in accordance with the observed higher adipogenic potential of iwatMSCs [41,42]. Furthermore, Baek et al. reported about an elevated expression of the secreted protein Galectin-3 (LGALS3) in obesity and T2D. The knockdown of Galectin-3 in 3T3-L1 cells resulted in a reduced adipocyte differentiation and an elevated expression of the adipogenesis mediators *Pparg* and *Cebps*, implicating its crucial role in adipogenic differentiation [43]. Even though a few of the depicted protein-coding genes are also linked to pro-fibrogenic properties (*Lgals3* [44], *Thbs2* [45]), these effects seem to be counter-regulated by genes with anti-fibrogenic potential (*Mmp19* [46], *Fst* [47]), which is in line with our functional analyses.

The ten secreted proteins that were predicted to be encoded by genes enriched in pMSCs, including *Mgp* (matrix gla protein) [48], *Igfbp4* (insulin-like growth factor binding protein 4) [49], and *Igfbp7* [50], depicted candidates which associate with pro-adipogenic events. This mirrors the in-principle capacity of pMSCs to develop into fat cells, as also observed in our experimental data. However, future studies have to be performed to measure the putatively secreted proteins in the supernatant of pMSCs and iwatMSCs. Interestingly, several prominent pro-fibrogenic proteins were predicted to be enriched in pMSCs, such as *Col3a1* (collagen type III alpha 1 chain) [42], *Cxcl14* (C-X-C motif chemokine 14) [51], *Angptl2* [52], *Slit3* (slit guidance ligand 3) [53], and *Smoc2* (sparc-related modular calcium binding protein-2) [54]. One of the leading fibrogenesis markers, COL3A1, which is regulated by TGF-β, was predicted to be secreted by pMSCs in our secretome analysis [42]. In addition, it is known that ANGTPL2, also regulated by TGF-β, has an inflammatory and fibrogenic potential [52] and that suppression of SMOC2 ameliorates kidney, pulmonary, and liver fibrosis [54,55,56]. 

The limitation of our study is the fact that several parts are based on in-silico analysis which only allows speculations. Thus, future studies are needed to confirm our results and to clarify which factors induce an elevated adipogenesis in pMSCs.

## 4. Conclusions

In conclusion, we show that pMSCs and iwatMSCs markedly differ in their transcriptome and cell lineage commitments. pMSCs displayed a higher fibrogenic, and iwatMSCs a higher adipogenic differentiation capacity in both, control and diabetes-susceptible mice. Thus, elevated pancreas adipogenesis in diabetes-prone animals has to be mediated by other factors, which require further validation in future experiments. Our results may contribute to an improved understanding how pancreatic fat contributes to the development of T2D as discussed recently [57].

## 5. Materials and Methods

### 5.1. Animals

Male C57BL/6J and NZO/HIBomDIfE mice from our own breeding (German Institute of Human Nutrition Potsdam-Rehbruecke, Nuthetal, Germany) were kept at 20 ± 2 °C on a 12:12 h light-dark cycle with ad libitum access to drinking water and a standard diet (V1534-300, ssniff, Soest, Germany). At the age of 8 weeks, non-fasted mice were sacrificed by cervical dislocation for preparation of iWAT and pancreas.

All mice were maintained in accordance with the National Institutes of Health guidelines for the care and use of laboratory animals. All treatments were approved by the ethics committee of the State Office of Environment, Health and Consumer Protection (Federal State of Brandenburg, Potsdam, Germany). The study is reported in accordance with the ARRIVE guidelines.

### 5.2. Isolation of Mesenchymal Stromal Cells with Adipogenic Potential

Flow cytometric isolation was performed based on procedures described previously by Quiclet et al. [58], with minor modifications. In brief, minced iWAT or pancreas from C57BL6/J male and NZO/HIBomDIfE were digested at 37 °C for 1 h in 2 mg/mL (iWAT) and 1 mg/mL (pancreas) of collagenase type II (Worthington Biochemical Corporation, Lakewood, NJ, USA), and samples were centrifuged at 290× *g* and 100× *g*, respectively. Cell suspensions were filtered through 40 µm strainers to remove doublets, large debris, and cell aggregates, and washed with sorting media (HBSS + 2% FBS). Cells were then stained with antibodies against PTPRC (CD45), PECAM (CD31), and LY6A (SCA1) as described before [12]. To achieve accurate sorting, Sca-1+ cells were marked with an APC (λe = 561 nm) and CD31+ and CD45+ cells with a FITC fluorophore (λe = 488 nm). Subsequently, CD45-:CD31-:SCA1+ cells were sorted for expression of SCA1 (positive selection) and CD45/CD31 (negative selection) to segregate mesenchymal stromal cells (MSCs) (Figure 1A). The labelling with fluorescence dyes enabled the segregation of MSCs from other cellular fractions, accounting for ~1 × 104 MSCs per mouse and ~15% of all isolated cells in the pancreas and for ~1 × 105 MSCs and ~50% of all isolated cells in the iWAT (Figure 1B,C). MSCs from the iWAT (iwatMSCs) and pancreas (pMSCs) were sorted using either a FACSAria III Cell Sorter (BD Biosciences, Franklin Lakes, NJ, USA) or a FACSMelody Cell Sorter (BD Biosciences, Franklin Lakes, NJ, USA). Next, pMSCs as well as iwatMSCs from three to five mice were pooled and cultured at an initial seeding density of 30,000 cells per well in 24-well plates. Finally, obtained cells were expanded for 4 days and isolated RNA was used for miRNome and transcriptome analysis of pMSCs and iwatMSCs (Figure 1A,D) as well as for adipogenic or fibrogenic differentiation.

### 5.3. Adipogenic Differentiation and Oil Red O Staining

Expanded cells were detached and re-seeded for differentiation in 48-well plates at a density of 15,000 cells per well. Adipogenesis was induced with an induction medium (DMEM 4.5% glucose, 10% FBS, 1% penicillin/streptomycin, 5 µg/mL insulin, 125 µM indomethacin, 0.5 mM IBMX, 5 µM dexamethasone, 1 nM T3) for 2 days and then differentiated for seven additional days with a differentiation medium (DMEM 4.5% glucose, 10% FBS, 1% penicillin/streptomycin, 5 µg/mL insulin, 1 nM T3, 1 µM Rosiglitazone). For quantification of lipid droplets, Oil Red O staining was performed as previously described [5]. Briefly, cells were washed twice with PBS and fixated with 4% formalin (Roti-Histofix, Carl Roth, Karlsruhe, Germany) for 1 h. Oil Red O stock solution was prepared by mixing 0.5 g Oil Red O in 100 mL isopropanol and a working solution by adding 6 mL of stock solution to 4 mL of dH2O and filtering through filter paper (Whatman #1, Cytiva Life Sciences, Buckinghamshire, UK). Total of 1 mL of working solution was added per well, and incubated for 1 h in the dark. Next, working solution was removed and cells were washed multiple times with dH2O to remove excess stain. Cell staining was analyzed and imaged in an Eclipse Ts1000 microscope (Nikon, Düsseldorf, Germany), plates were dried at RT and Oil Red O was extracted with isopropanol (220 µL per well). About 100 µL of extracted solution was transferred to a clear 96-well plate, absorbance measured at 562 nm in a Synergy H1 Hybrid Multi-mode Reader spectrophotometer (BioTek/Agilent Technologies, Santa Clara, USA) and calculated using a standard curve.

### 5.4. Fibrogenic Differentiation and Immunocytochemistry

Fibrogenesis of expanded cells (15,000 cells per well) was carried out by incubating cells in a fibrogenic medium (DMEM 1% glucose, 10% FBS, 1% penicillin/streptomycin, 1 ng/mL TGF-β1) for 5 days.

For immunocytochemistry, cells were fixated with 4% formalin (Roti-Histofix, Carl Roth, Karlsruhe, Germany) for 1 h. Cells were then permeabilized using Triton X-100 (0.1% in PBS), blocked with BSA (3% in PBS) and incubated with the primary antibody against αSMA (Sigma-Aldrich # A5228; St. Louis, Missouri, USA) overnight. Next day, cells were incubated with secondary antibody (Abcam # ab150116; Berlin, Germany) for 1 h in dark. Cells were then washed with PBS and incubated with DAPI (300 nM) for 5 min. Staining was visualized in a BZ900 Fluorescence Microscope (Keyence, Neu-Isenburg, Germany) and analyzed using ImageJ software.

### 5.5. RNA Isolation and Real-Time qPCR

RNA of differentiated MSCs was isolated (miRNeasy Micro Kit, Qiagen, Hilden, Germany) and reversed transcribed (M-MVL RT, Promega, Madison, WI, USA) for quantitative real-time PCR. Genes of interest were detected using specific TaqMan (Thermo Fisher Scientific, Waltham, MA, USA) and IDT (Integrated DNA Technologies, Coralville, IA, USA) probes. Expression levels were evaluated using the 2(-Delta CT) method [4] with eukaryotic translation elongator factor 2 (*Eef2*) as internal control. 

For RNA sequencing, isolated MSCs were expanded for 4 days and RNA was isolated according to the manufacturer’s protocol (miRNeasy Micro Kit, Qiagen, Hilden, Germany).

### 5.6. RNA and Small RNA Sequencing

Total RNA was mRNA enriched, fragmented, and underwent DNA nanoball synthesis to get sequenced on the BGISeq platform (Yantian District, Shenzhen, China). After transcriptome sequencing, adapters and low-quality reads were filtered and FastQC v0.11.8 was conducted to check quality of the samples. Next, reads were aligned to the reference genome (GRCm38.p6) using STAR v.2.7.0f, and FPKM values for transcripts were determined by STRINGTIE v1.3.6, both with default options for paired reads. 

For small RNA sequencing, small RNA enrichment, purification steps, multiple adapter ligation, and amplification steps were performed. Next, unique molecular identifiers (UMI) were synthesized to the first strand and pooling cyclization was performed. The final library got sequenced using BGISEQ. BGI delivered all bioinformatic steps including raw data, small RNA identification, classification to miRNA, siRNA and piRNA as well as the determined expression values in TPM (transcripts per million).

### 5.7. Bioinformatic Analyses

Differentially expressed transcript and miRNA analysis were performed using R (v4.0.2). Transcripts and miRNAs were filtered for mean expression values below 1 FPKM (fragments per kilobase per million) and 1 TPM, respectively. *p*-values were calculated using Welch’s *t*-test. For further analysis of the miRNome, only mature miRNA from the smallRNA dataset was used. In this dataset, one outlier sample was discovered and excluded. Volcano plots were generated via the EnhancedVolcano package (v.1.6.0). The RDAVIDWebService package (v1.26.0) was utilized to perform the GO analysis and GOplot (v1.0.2). Additional pathway analyses were performed using Ingenuity Pathway Analysis (IPA, Qiagen Silicon Valley, Redwood City, CA, USA). The R-package ComplexHeatmap (v2.4.3) was used to create heatmaps.

### 5.8. miRNA Target Prediction

The miRNA target gene prediction was performed as previously described [32]. In brief, target genes were determined using three databases (DIANA-TarBase, miRecords, and miRTarBase [59,60,61]), which list experimentally validated miRNA-mRNA interactions (detected by HITS-CLIP [62], luciferase reporter or in vitro assays).

### 5.9. Statistics and Plotting

As pancreatic stellate cells and pMSCs have common developmental origins and share several similar characteristics in terms of gene expression and functionalities [63,64], we compared the transcriptome of pMSCs with single cell RNA sequencing (scRNAseq) data from mouse pancreatic stellate cells. The comparison revealed a high enrichment of stellate cell marker genes among the top 1000 expressed pMSC genes (odds-ratio:10.9 p-val: 2.2e-16) (Appendix A).

Statistical significance was analyzed using Student’s t test or one-way ANOVA by comparing the test groups with the appropriate control groups. Data are presented as mean values ± SEM. Statistical significance is expressed as * <0.05, ** <0.01, and *** <0.001. Where applicable, the number of replicates (*n*) was stated in the figure legend. Plots were created using GraphPad 9.0 (GraphPad Software, San Diego, CA, USA).

## Figures and Tables

**Figure 1 ijms-23-02108-f001:**
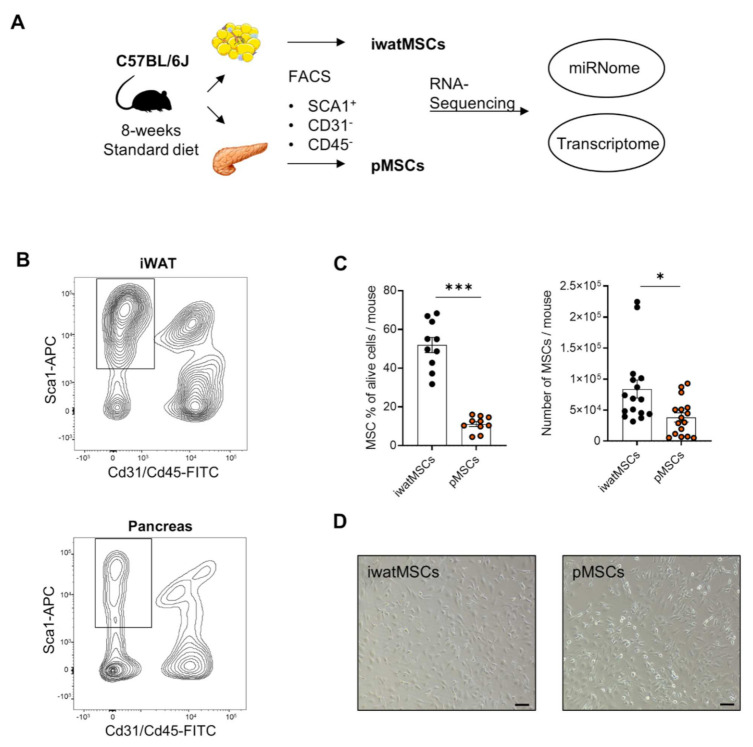
Isolation of mesenchymal stromal cells from inguinal white adipose tissue and pancreas for transcriptome and miRNome analyses. (**A**) Study design. iWAT and pancreas were collected from the same C57BL/6J mice and MSCs were isolated via fluorescence-activated cell sorting (FACS). After expansion, iwatMSCs and pMSCs were harvested and transcriptome and miRNome analyses were performed. (**B**) FACS of cells isolated from iWAT and pancreas. Box marks the non-endothelial (CD31-), non-hematopoietic (CD45-), adipogenic progenitor (SCA1+) MSC subpopulation. (**C**) Graphical representation of alive iwatMSC (black) and pMSC (orange) fraction per mouse and the total sorted iwatMSC and pMSC number per mouse. (**D**) Representative microscopic images of iwatMSCs and pMSCs after 4 days of expansion on cell culture plates. Scale bars: 100 µm. Differential expression defined by Student’s *t*-test with Welch’s correction unadjusted * *p* < 0.05, *** *p* < 0.0001.

**Figure 2 ijms-23-02108-f002:**
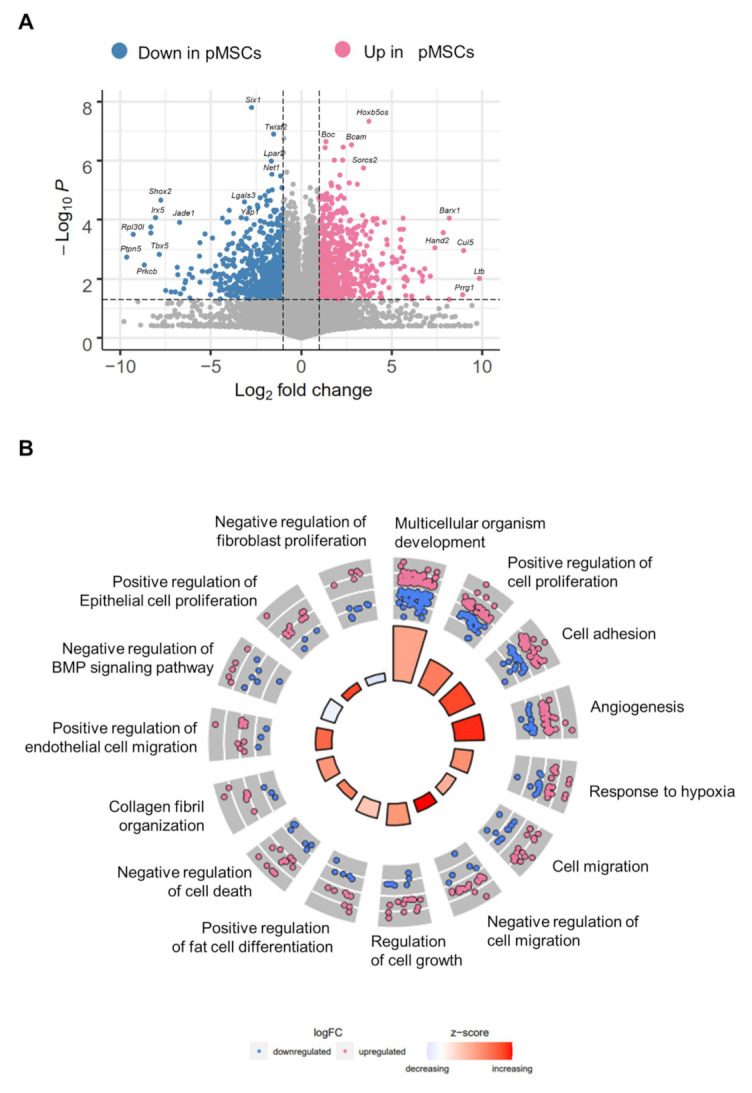
Comparative transcriptome and pathway analyses show various distinct biological processes between pMSCs and iwatMSCs. (**A**) Volcano plot of 26,735 expressed transcripts in iwatMSCs and pMSCs. Blue and pink dots depict up- and down-regulated transcripts in pMSCs, respectively (*n* = 8; Student’s *t*-test with Welch’s correction unadjusted *p* < 0.05). (**B**) Circular visualization of gene-ontology enrichment analysis (GOcircle) with 1227 genes that differ in their expression between iwatMSCs and pMSCs. The inner circle indicates the main biological processes to be either increased (red) or decreased (blue) in pMSCs (shown by the z-score). The outer circle depicts scatter plots for genes allocated to the most significant biological processes and the direction of regulation in pMSCs. Blue dots mark down-regulated and red dots up-regulated genes in pMSCs. (**C**) Gene expression network identified by Ingenuity Pathway Analysis (IPA). The network links 11 of the DEGs to genes known for cell-growth and adipogenesis. Down-regulated genes in pMSCs are shown in green and up-regulated genes are marked in red.

**Figure 3 ijms-23-02108-f003:**
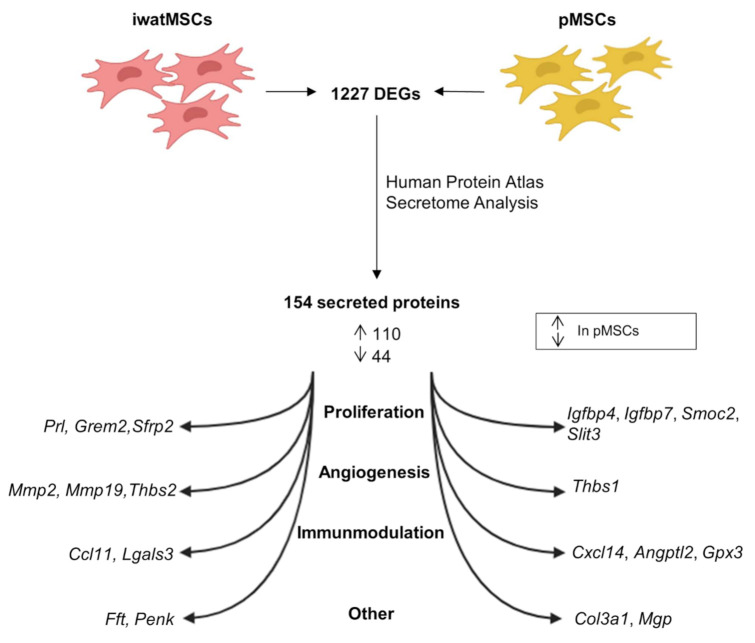
In-silico secretome analysis predicts a higher number of secreted proteins in pMSCs compared to iwatMSCs. The 1227 DEGs between iwatMSCs and pMSCs MSCs were used to identify putatively secreted proteins via Human Protein Atlas. The in-silico secretome analysis is based on expression patterns and predicted 154 secreted proteins among the DEGs, of which 110 might be secreted by pMSCs and 44 by iwatMSCs. Shown are top-ten protein-coding genes predicted to be mainly secreted by iwatMSCs and pMSCs and their putative function, respectively.

**Figure 4 ijms-23-02108-f004:**
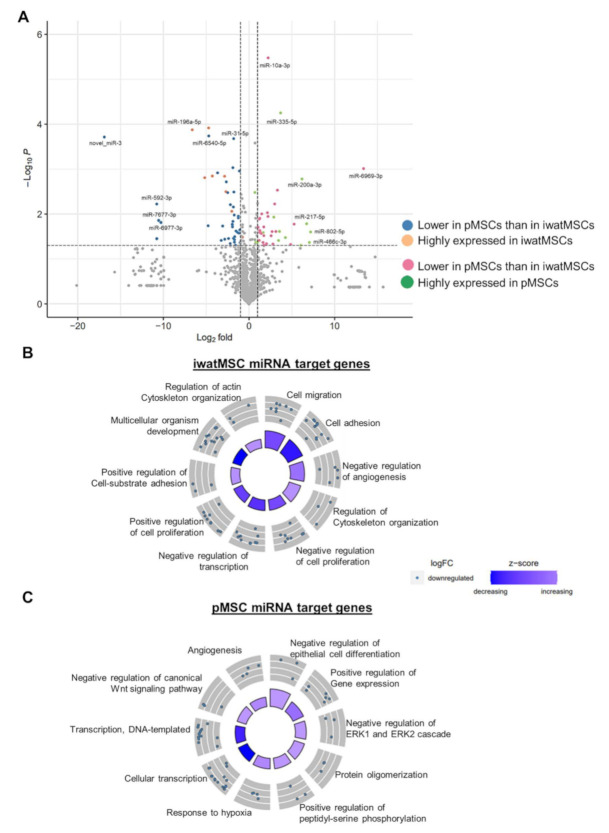
Differentially expressed miRNA between iwatMSCs (iWAT) and pMSCs (PANC) might participate in their distinct expression profiles. (**A**) Volcano plot of 1240 expressed miRNAs showing differences between iwatMSCs versus pMSCs. 66 miRNAs were higher expressed (blue and orange dots), while 55 were lower expressed (pink and green dots) in pMSCs than in iwatMSCs. Green dots highlight miRNA which are predominantly expressed in iwatMSCs, whereas orange dots show miRNA mainly regulated in pMSCs (defined by *p* ≤ 0.05 and log2FC ≥ 2). (*n* = 8; Student’s *t*-test with Welch’s correction unadjusted *p* < 0.05). (**B**,**C**) Gene ontology (GO) enrichment analyses with predicted target genes of highly expressed miRNAs in iwatMSCs and pMSCs (defined by *p* ≤ 0.05 and log2FC ≥ 2), respectively. The GOcircle plots show the top 10 enriched GO-terms in the category biological processes. The inner circle reflects the direction of the indicated biological processes (dark blue = increased, light blue = decreased). The outer circle depicts scatter plots for miRNA target genes allocated to the respective biological processes. (**D**,**E**) miRNA nearly exclusively expressed in iwatMSCs and pMSCs, respectively. Target genes with a repression of log2FC > 1.5 are shown accordingly in boxes (* *p* < 0.05, ** *p* < 0.005, *** *p* < 0.0001).

**Figure 5 ijms-23-02108-f005:**
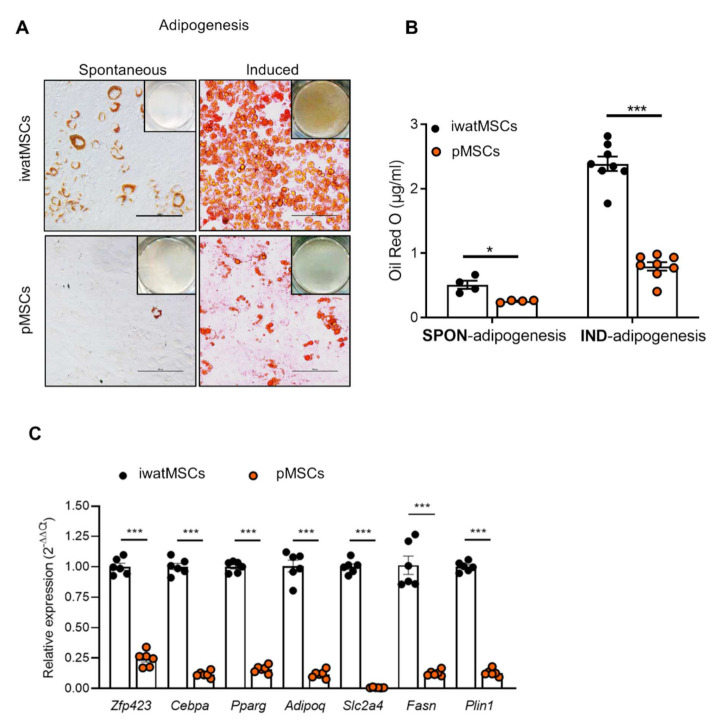
pMSCs display a significantly reduced adipogenesis in comparison to iwatMSCs. After flow-cytometric sorting, iwatMSCs and pMSCs were seeded on cell culture plates and expanded for 4 days. After an adipogenic induction of 2 days, cells were differentiated for another 10 days before analysis. (**A**) Representative images of Oil Red O stainings in iwatMSCs and pMSCs showing spontaneous (SPON-) and induced (IND-) adipogenesis. Scale bars: 100 µm. (**B**) Quantification of Oil Red O staining (*n* = 4–8 per group). (**C**) Relative expression of indicated adipogenesis markers in iwatMSCs and pMSCs after treatment with the differentiation cocktail (*n* = 5–6 per group). Data shown as mean ± SEM. Statistical difference was calculated using Student’s *t*-test with Welch’s correction (* *p* < 0.05, *** *p* < 0.001).

**Figure 6 ijms-23-02108-f006:**
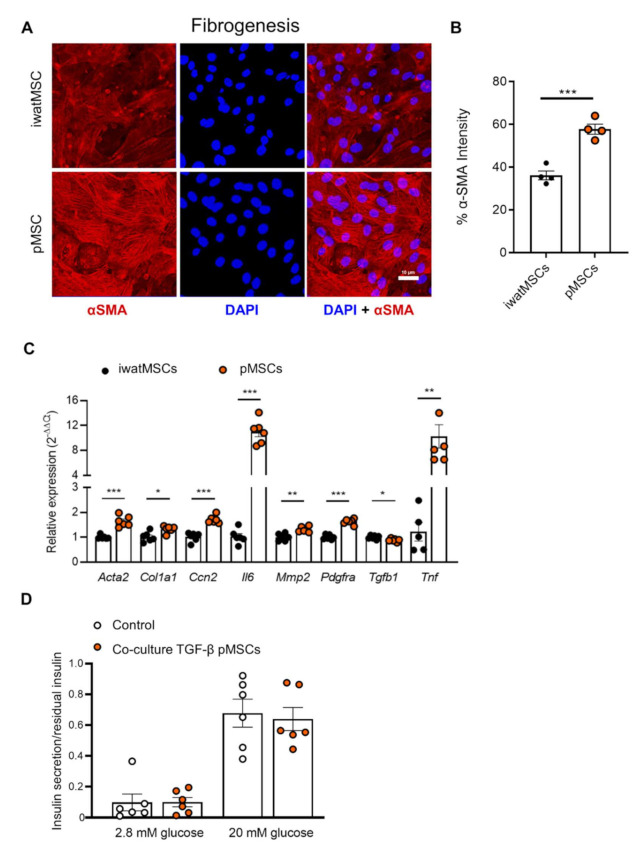
pMSCs exhibit increased levels of the fibrosis-marker α-SMA. Sorted and expanded (4 days) pMSCs and iwatMSCs were treated with TGF-β1 for 5 days to induce fibrogenesis. (**A**) Representative images of iwatMSCs and pMSCs stained for the fibrogenic marker α-SMA (red). Nuclei were detected with DAPI (blue). DAPI: 1200 per group, scale bars: 40 µm. (**B**) Morphometric analysis of α-SMA signal intensity (*n* = 4–8 per group). (**C**) Relative expression levels of indicated fibrogenesis markers in iwatMSCs and pMSCs after TGF-β1 treatment (*n* = 6 per group). (**D**) Glucose-stimulated insulin secretion of pancreatic islets of B6 mice alone or co-cultured with TGF-β1 treated pMSCs. * *p* < 0.05, ** *p* < 0.005, *** *p* < 0.001.

**Figure 7 ijms-23-02108-f007:**
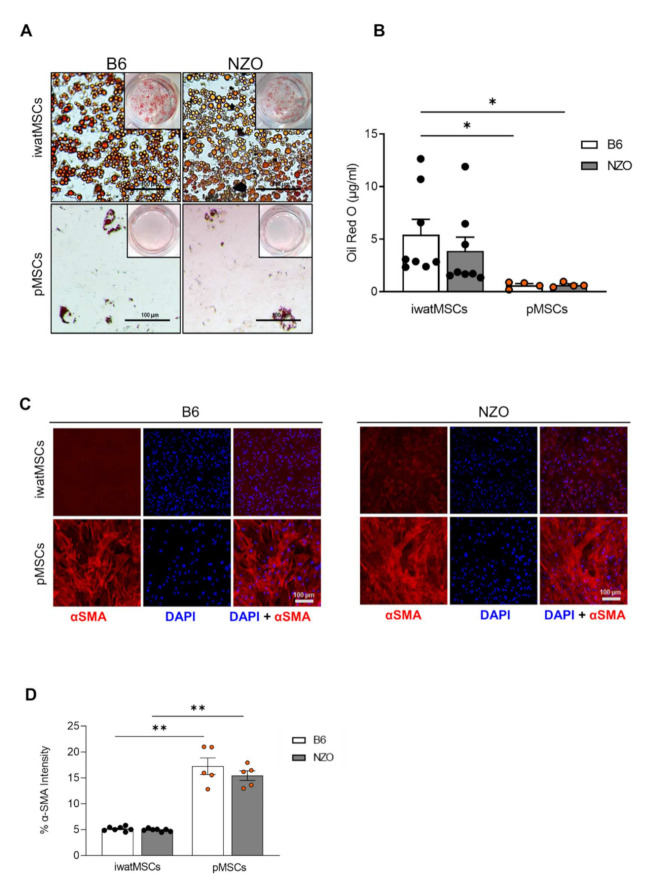
pMSCs and iwatMSCs of diabetes-susceptible NZO mice display similar lineage commitments as shown for B6 animals. (**A**) After flow-cytometric sorting, iwatMSCs and pMSCs of B6 and NZO mice were seeded on cell culture plates and expanded for 4 days. After an adipogenic induction of 2 days, cells were differentiated for another 10 days before analysis. Representative images of Oil Red O stainings in iwatMSCs and pMSCs of both mouse strains showing induced adipogenesis. Scale bars: 100 µm. (**B**) Quantification of Oil Red O stainings (*n* = 5 per group). (**C**) For analysis of the fibrogenic differentiation capacity, iwatMSCs and pMSCs were treated with TGF-β1 for 5 days. Representative images of iwatMSCs and pMSCs of B6 and NZO mice stained for the fibrogenic marker α-SMA (red). Nuclei were detected with DAPI (blue). DAPI: 1200 per group, scale bars: 40 µm. (**D**) Morphometric analysis of α-SMA signal intensity (*n* = 5 per group). Data shown as mean ± SEM. Statistical difference was calculated using Student’s *t*-test with Welch’s correction (* *p* < 0.01, ** *p* < 0.05).

## Data Availability

All data used in this manuscript are available upon request. Accession numbers for raw RNA-seq and miRNA data are GEO: GSE182243.

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
