# Peer review of "Distinct Adipogenic and Fibrogenic Differentiation Capacities of Mesenchymal Stromal Cells from Pancreas and White Adipose Tissue"

_ijms, 2022, doi:10.3390/ijms23042108_

Round 1
Reviewer 1 Report
The present study demonstrated that development of pancreatic steatosis was not related to cell-autonomous transcriptomic changes and miRNA-based signals. I like to give the following comments.
- In the abstract, MCSs seems the error of MSCs. Please check it.
- Compared the MSCs isolated from pancreas (pMSCs) and iWAT (iwatMSCs) of B6 mice, iwatMSCs exhibit differentiation into adipocytes significantly more than pMSCs. Is this also observed in diabetes-susceptible NZO mice? Please describe it in discussion.
- Figure 7 showed the same commitments of both MSCs from B6 and NZO mice. Please compare that B6 mice seem more resistant to insulin resistance in a wat varied with NZO mice. How is the body weight of NZO mice?
- In the discussion, a fibrogenesis-favoring cell fate of pMSCs that needs to explain in clear.
- The 10 secreted proteins were not determined in current study. Why?
- In conclusion, elevated pancreas adipogenesis in diabetes-prone animals that needs the reference(s) to support.
- Limitation of current study seems ignored.
Author Response
- In the abstract, MCSs seems the error of MSCs. Please check it.
We thank reviewer 1 for indicating this error which we corrected.
- Compared the MSCs isolated from pancreas (pMSCs) and iWAT (iwatMSCs) of B6 mice, iwatMSCs exhibit differentiation into adipocytes significantly more than pMSCs. Is this also observed in diabetes-susceptible NZO mice? Please describe it in discussion.
The reviewer is right, pancreatic MSCs from B6 mice exhibit a lower capacity to differentiate into adipocytes than MSCs form iWAT (shown in Figure 5). In Figure 7 we demonstrated that pancreas (pMSCs) and iWAT (iwatMSCs) of the diabetes-susceptible mice behave very similar. pMSCs of NZO mice differentiate to a much lower degree to adipocytes than their iwatMSCs. We discussed this finding in more detail in the revised manuscript and stated in the first paragraph of the discussion: “Interestingly, also pMSCs of diabetes-susceptible NZO mice, in which we previously detected a high number of adipocytes in pancreas [12], exhibit a low degree to differentiate into adipocytes… Thus, we hypothesize that other species-specific factors are responsible for the adipocyte accumulation in pancreas.”
In addition, we included a last paragraph with a conclusion of the study in which we also picked up this aspect and stated: “Thus, elevated pancreas adipogenesis in diabetes-prone animals has to be mediated by other factors, which requires further validation in future experiments.”
- Figure 7 showed the same commitments of both MSCs from B6 and NZO mice. Please compare that B6 mice seem more resistant to insulin resistance in a wat varied with NZO mice. How is the body weight of NZO mice?
As mentioned in the response above, the individual capacities of pMSCs and iwatMSCs to differentiate into adipocytes is similar between B6 and NZO. This appears even to be independent of their whole body and adipose tissue insulin sensitivity. The mean body weight of B6 mice used in this study was 24.4 g and of NZO 34.9 g.
- In the discussion, a fibrogenesis-favoring cell fate of pMSCs that needs to explain in clear.
The expression profile of the pMSCs indicated their elevated fibrogenic cell fate. Therefore, we performed in vitro-analysis and tested the reaction of pMSCs to an activator of fibrogenesis, TGF-β. As demonstrated in Figure 5, pMSCs showed a higher degree of fibrogenesis and elevated expression of fibrogenic marker genes than iwatMSCs. This means that the pMSCs exhibit the capacity to induce pancreas fibrosis. We referred to this aspect in the discussion and stated: “In fact, our in vitro studies showed the differentiation of pMSCs to fibrogenic cells which might indicate that pMSCs play a role in fibrosis of the pancreas, e.g. in the development of pancreatitis.”
- The 10 secreted proteins were not determined in current study. Why?
We are sorry for not measuring the proteins. The data shown in Figure 3 are results of an in silico-analysis which should only give an idea how pMScs and iwatMSCs might communicate with other pancreatic or adipose tissue cells, respectively. The aim of the actual study was to provide information about the differences in the mRNA and microRNA expression profile and to identify some general differences between the two types of MSCs. Future studies are required to investigate if specific secreted proteins are indeed released by iwatMSCs and pMSCs and which impact these secreted proteins have to other cells. We referred to this in the discussion and stated: “However, future studies have to be performed to measure the putatively secreted proteins in the supernatant of pMSCs and iwatMSCs.”
- In conclusion, elevated pancreas adipogenesis in diabetes-prone animals that needs the reference(s) to support.
Besides the diabetes-prone NZO mouse model there is only the KK‐Ay mouse, that is known to develop fat infiltrations in the pancreas. In fact, KK‐Ay mice are also a T2D mouse model. To support our hypothesis for elevated pancreas adipogenesis in diabetes-prone animals, we added the reference in which the fat infiltrations in KK‐Ay mice are shown [line 331-336 (https://doi.org/10.1111/cas.13766)]. Furthermore, we included the link between pancreatic fat, impaired glucose metabolism and T2D in human by citing a review in which the possible causes and metabolic consequences were summarized recently (Wagner, R., et al., Metabolic implications of pancreatic fat accumulation. Nat Rev Endocrinol, 2022. 18(1): p. 43-54.)
- Limitation of current study seems ignored.
We are sorry for not mentioning the limitations of our study. We now included the following sentence at the end of the discussion: “The limitation of our study is the fact that several parts are based on in silico-analysis which only allow speculations. Thus, future studies are needed to confirm our results and to clarify which factors induce an elevated adipogenesis in pMSCs.”

Reviewer 2 Report
The manuscript entitled Distinct adipogenic and fibrogenic differentiation capacities of
mesenchymal stromal cells from pancreas and white adipose tissue has been submitted by Heja Aga et al. to the International Journal of Molecular Sciences. The manuscript documents an impressive scientific work which is suited to be published in IJMS.
I do not understand this sentence in the abstract. „Our previous studies have shown that diabetes-susceptible mice accumulate more adipocytes in the pancreas than diabetes-resistant mice and the co-culture of pancreatic islets and adipocytes affect insulin secretion.“ ???
What do the authors mean?
Our previous studies have shown that diabetes-susceptible mice accumulate more adipocytes in the pancreas than diabetes-resistant mice. (and) This is also the case for the co-culture of pancreatic islets and adipocytes affect insulin secretion. ???
Line 189: …… ganglioside (glycosphingolipid) sialylation 189 [17]. Gangliosides are by definition sialylated glycolipids. What do the authors mean? A ganglioside e.g. GM1 becomes by sialylation e.g. GD1a or a glycolipid without any sialic acid residue becomes a ganglioside?
The conclusion part is rather short.
It could be beneficial for the manuscript when the author will correlate their findings with these three publications.
Expansion and inflammation of white adipose tissue - .....
Wenjing Liu, Dahui Li, Handi Cao, Haoyun Li und Yu Wang
Biological Chemistry. https://doi.org/10.1515/hsz-2019-0451
Contribution of adipogenesis to healthy adipose tissue expansion in obesity.
Lavanya Vishvanath, Rana K. Gupta
J. Clin. Invest. 2019;129(10): 4022-4031. https://doi.org/10.1172/JCI129191.
Atrial fibrillation and cardiac fibrosis.
Sohns C, Marrouche NF. Eur Heart J. 2020 Mar 7;41(10):1123-1131.
doi: 10.1093/eurheartj/ehz786.
Author Response
The manuscript entitled Distinct adipogenic and fibrogenic differentiation capacities of mesenchymal stromal cells from pancreas and white adipose tissue has been submitted by Heja Aga et al. to the International Journal of Molecular Sciences. The manuscript documents an impressive scientific work which is suited to be published in IJMS.
We thank reviewer 2 for this positive feedback.
I do not understand this sentence in the abstract. „Our previous studies have shown that diabetes-susceptible mice accumulate more adipocytes in the pancreas than diabetes-resistant mice and the co-culture of pancreatic islets and adipocytes affect insulin secretion.“ ???
What do the authors mean?
Our previous studies have shown that diabetes-susceptible mice accumulate more adipocytes in the pancreas than diabetes-resistant mice. (and) This is also the case for the co-culture of pancreatic islets and adipocytes affect insulin secretion. ???
We rephrased this part in the abstract as follows: “Our previous studies have shown that diabetes-susceptible mice accumulate more adipocytes in the pancreas than diabetes-resistant mice. In addition, we have demonstrated that the co-culture of pancreatic islets and adipocytes affect insulin secretion.”
Line 189: …… ganglioside (glycosphingolipid) sialylation 189 [19]. Gangliosides are by definition sialylated glycolipids. What do the authors mean? A ganglioside e.g. GM1 becomes by sialylation e.g. GD1a or a glycolipid without any sialic acid residue becomes a ganglioside?
We are sorry for this confusing statement. We corrected the sentence and wrote: “Deletion of this gene induces late onset of obesity and insulin resistance, particularly in adipose tissue, which was associated with altered ganglioside profiles [19].”
The conclusion part is rather short.
We thank reviewer 2 for this comment and agree that the paper lacked a conclusion at the end. We added a paragraph referring to the limitation of our study as well as a paragraph with a conclusion at the end of the discussion: “The limitation of our study is the fact that several data are based on in-silico analysis which only allow speculations. Thus, future studies are needed to confirm our results and to clarify which factors induce an elevated adipogenesis in pMSCs.
In conclusion, we show that pMSCs and iwatMSCs markedly differ in their transcriptome and cell lineage commitments. pMSCs displayed a higher fibrogenic and iwatMSCs a higher adipogenic differentiation capacity in both, control and diabetes-susceptible mice. Thus, elevated pancreas adipogenesis in diabetes-prone animals has to be mediated by other factors, which requires further validation in future experiments. Our results may contribute to an improved understanding how pancreatic fat contributes to the development of T2D as discussed recently [57].”
It could be beneficial for the manuscript when the author will correlate their findings with these three publications.
Expansion and inflammation of white adipose tissue - .....
Wenjing Liu, Dahui Li, Handi Cao, Haoyun Li und Yu Wang
Biological Chemistry. https://doi.org/10.1515/hsz-2019-0451
We thank reviewer 2 for this recommendation. This publication was referred to in lines 46-48 and 310-311.
Contribution of adipogenesis to healthy adipose tissue expansion in obesity.
Lavanya Vishvanath, Rana K. Gupta
- Clin. Invest. 2019;129(10): 4022-4031. https://doi.org/10.1172/JCI129191.
We thank reviewer 2 for this recommendation. This publication was referred to in lines 48-52 and 219-222.
Atrial fibrillation and cardiac fibrosis.
Sohns C, Marrouche NF. Eur Heart J. 2020 Mar 7;41(10):1123-1131.
doi: 10.1093/eurheartj/ehz786.
Unfortunately, it was not possible for us to refer to this publication in our manuscript, as the mentioned paper was very specifically about atrial fibrillation and cardiac fibrosis, which we do not address in our research.
